# Children’s Greenness Exposure and IQ-Associated DNA Methylation: A Prospective Cohort Study

**DOI:** 10.3390/ijerph18147429

**Published:** 2021-07-12

**Authors:** Kyung-Shin Lee, Yoon-Jung Choi, Jin-Woo Cho, Sung-Ji Moon, Youn-Hee Lim, Johanna-Inhyang Kim, Young-Ah Lee, Choong-Ho Shin, Bung-Nyun Kim, Yun-Chul Hong

**Affiliations:** 1Department of Preventive Medicine, Seoul National University College of Medicine, Seoul 03080, Korea; kslee0116@snu.ac.kr (K.-S.L.); pastelorange2012@gmail.com (Y.-J.C.); kajaman3@snu.ac.kr (S.-J.M.); limyounhee@gmail.com (Y.-H.L.); 2Environmental Health Center, Seoul National University College of Medicine, Seoul 03080, Korea; 3Department of Statistics, University of Pittsburgh, Pittsburgh, PA 15260, USA; kevinjwcho@pitt.edu; 4Section of Environmental Health, Department of Public Health, University of Copenhagen, 1014 Copenhagen, Denmark; 5Department of Psychiatry, Hanyang University Medical Center, Seoul 04763, Korea; iambabyvox@hanmail.net; 6Department of Pediatrics, Seoul National University Children’s Hospital, Seoul National University College of Medicine, Seoul 03080, Korea; nina337@snu.ac.kr (Y.-A.L.); chshinpd@snu.ac.kr (C.-H.S.); 7Division of Children and Adolescent Psychiatry, Department of Psychiatry, Seoul National University Hospital, Seoul 03080, Korea; 8Institute of Environmental Medicine, Seoul National University Medical Research Center, Seoul 03080, Korea

**Keywords:** greenness, epigenetics, DNA methylation, intelligence quotient, cytosine-guanine dinucleotide sites

## Abstract

Epigenetics is known to be involved in regulatory pathways through which greenness exposure influences child development and health. We aimed to investigate the associations between residential surrounding greenness and DNA methylation changes in children, and further assessed the association between DNA methylation and children’s intelligence quotient (IQ) in a prospective cohort study. We identified cytosine-guanine dinucleotide sites (CpGs) associated with cognitive abilities from epigenome- and genome-wide association studies through a systematic literature review for candidate gene analysis. We estimated the residential surrounding greenness at age 2 using a geographic information system. DNA methylation was analyzed from whole blood using the HumanMethylationEPIC array in 59 children at age 2. We analyzed the association between greenness exposure and DNA methylation at age 2 at the selected CpGs using multivariable linear regression. We further investigated the relationship between DNA methylation and children’s IQ. We identified 8743 CpGs associated with cognitive ability based on the literature review. Among these CpGs, we found that 25 CpGs were significantly associated with greenness exposure at age 2, including cg26269038 (Bonferroni-corrected *p* ≤ 0.05) located in the body of *SLC6A3*, which encodes a dopamine transporter. DNA methylation at cg26269038 at age 2 was significantly associated with children’s performance IQ at age 6. Exposure to surrounding greenness was associated with cognitive ability-related DNA methylation changes, which was also associated with children’s IQ. Further studies are warranted to clarify the epigenetic pathways linking greenness exposure and neurocognitive function.

## 1. Introduction

Exposure to greenness in urban areas is estimated to have physiological and psychosocial health benefits in children [1,2,3,4,5]. Urban greenness can contribute to reducing the harmful effects of urbanization [6], including reducing exposure to environmental toxicants [7] and noise [8], increasing the level of physical activity [9,10], and enhancing social cohesion [11] in children. However, the biological mechanisms underlying the association between greenness exposure and desirable health effects remain unclear.

Many epidemiological studies have shown that an adverse intrauterine environment, including smoking [12,13,14], chemical exposures [15,16,17,18], ambient air pollution [19,20], and stress [21,22], may result in epigenetic perturbations of the developing fetus and can be associated with an increased risk of adverse health outcomes in later life. Additionally, exposure to heavy metals in early childhood (ages 1–4 years) was significantly associated with epigenetic change such as H19 hypermethylation, which may contribute to growth and metabolic diseases [23]. Hence, DNA methylation may be a possible mechanism by which early life environmental factors contribute to an increased risk of diseases in later life [24,25]. In addition, epigenetic modifications, such as DNA methylation, are susceptible to genetic and environmental factors and may provide insights into individual differences in health outcomes [26]. Epigenetic change is hypothesized to be a regulatory pathway through which exposure to greenness in early childhood may influence child development and health.

However, few studies have assessed the association between greenness exposure and changes in DNA methylation. Xu et al. (2021) showed an association between greenness exposure and gene and their interactions on blood-derived DNA methylation in 479 adult females [27]. They found greenness-associated DNA methylation changes of cytosine-guanine dinucleotide (CpG) sites at genes related to various human diseases such as mental disorders, neoplasms, nutritional and metabolic diseases [27]. The *CNP* gene at cg04720477 was strongly associated with greenness exposure and encodes a protein that has been related to low expression in brain tissue of schizophrenic [28] and depressive patients [29]. These results suggest that high greenness may be related to elevated *CNP* expression due to reduced methylation of this gene in female adults [27]. However, this study had a cross-sectional design, and so it was unable to determine whether DNA methylation plays a role in the association between improved mental health and exposure to greenness. In addition, they did not estimate the epigenetic impact of greenness on clinical outcomes.

Previously, we found that residential greenness was associated with IQ in the Environment and Development of Children (EDC) cohort study of 189 children [30]. We suggest that postnatal greenness exposure is more strongly associated with IQ in children than prenatal exposure to greenness. Cognitive skills are a strong predictor of a wide range of later life outcomes [31]. We hypothesized that residential greenness in early childhood may be associated with epigenetic alterations and that these alterations may influence later childhood cognitive outcomes. Using a sub-study of 59 children with DNA methylation data, we sought to evaluate the association between residential greenness exposure and DNA methylation changes reported from genome-wide association studies (GWAS) and epigenome-wide association studies (EWAS) of cognitive ability in children in a hypothesis-driven approach. We then investigated the association between the DNA methylation changes, which were significantly associated with greenness exposure, and children’s IQ in the prospective EDC cohort.

## 2. Materials and Methods

### 2.1. Study Population

Our research was based on a subset of the EDC study cohort, an ongoing prospective cohort study designed to evaluate the association between prenatal and postnatal environmental exposures and physical or cognitive development. Detailed information on the study design has been described elsewhere [32]. Briefly, a total of 726 eligible pregnant women from eight local hospitals in Seoul and Gyeonggi province of South Korea were enrolled from August 2008 to July 2010. We collected urine and blood samples to estimate exposure to environmental factors during the second trimester of pregnancy. A total of 425 children aged 2 years and 574 children aged 6 years at enrolment were followed up. DNA methylation analysis was conducted in a sub-study of 59 participants using blood samples collected at the age of 2 years. The study protocol, including ethical approval and participant consent was reviewed and approved by the Institutional Review Board of the Seoul National University Hospital (IRB No. C-1201-010-392).

### 2.2. Systematic Review of Literature and Selection of Candidate Cytosine-Guanine Dinucleotide Sites

As we were specifically interested in the question of whether DNA methylation mediates the effects of exposure to greenness on children’s IQ, we targeted CpG sites that were more likely to be involved in cognitive ability instead of scanning the whole epigenome. For the selection of previous EWAS or GWAS on association with cognitive abilities, we searched PUBMED and EMBASE on April 1, 2021, using keywords (“epigenome-wide association study” or “genome-wide association study”) and (“intelligence” or “cognitive ability” or “cognitive development”) from titles or abstracts. The selection criteria were EWAS or GWAS regarding cognitive ability in healthy children or adults. From previous EWAS or GWAS that investigated the association between DNA methylation and cognitive ability in healthy children or adults, we identified CpG sites associated with cognitive-ability-related parameters (Figure 1). In the GWAS, single nucleotide polymorphisms (SNPs) associated with cognitive ability were identified, and then the genes annotated to these SNPs were identified. The CpG sites associated with these genes were pooled using the Database for Annotation, Visualization, and Integrated Discovery (DAVID, http://david.abcc.ncifcrf.gov/home.jsp). We added IQ-related genes from bibliographyies, in which the genes were identified through enrichment analyses.

### 2.3. Measurement of Residential Surrounding Greenness

To estimate exposure to greenness, the residential addresses were collected at the age of 2 years. The surrounding greenness was recorded using Landsat image data from the IKONOS satellite images [33] and Korean Arirang satellite images taken by the Environmental Geographic Information Service of the Ministry of the Environment (https://egis.me.go.kr/main.do). To estimate exposure to greenness, densities of greenness were calculated within buffer radii of 100, 500, 1000, 1500, and 2000 m of each child’s residential address. We then determined the percentage of greenness (density) from the area within each buffer radius. We separately analyzed two types of greenness, namely, natural greenness, forest or natural grassland, and built greenness, including artificial grassland, urban parks, and street trees. We did not analyze the effect of natural greenness within buffer radii of both 100 and 500 m because natural greenness was barely observed within these ranges.

### 2.4. Measurement of DNA Methylation

#### 2.4.1. Assessment of DNA Methylation at Age 2 in the Environment and the Development of Children Study Cohort

We performed genome-wide DNA methylation analyses using the whole blood samples of 59 2-year-old children as described in an earlier study [34]. Briefly, DNA samples were tested for quality using a NanoDrop^®^ ND-1000 UV-VIS spectrophotometer (Thermo Fisher Scientific, Wilmington, DE, USA). Electrophoresis was performed using 1% agarose gel, and samples with genomic DNA (gDNA) were diluted to 50 ng/μL based on Quanti-iT Picogreen quantification (Thermo Fisher Scientific, Wilmington, DE, USA). The gDNA samples (minimum 500 ng) were diluted, then bisulfite-converted using the Zymo EZ DNA methylation kit (Zymo Research, Irvine, CA, USA), and the DNA was then amplified to be used on the DNA BeadChip. At age 2, we used the Illumina Infinium Human MethylationEPIC BeadChip, which yielded 850,000 CpG sites (Illumina, San Diego, CA, USA). Images were read by the Illumina BidArray Reader, and the image intensities were extracted using the Illumina GenomeStudio software. Microarrays were handled by Macrogen Co. (Seoul, Korea). For functional annotation analysis, we used the DAVID (david.abcc.ncifcrf.gov) tool.

#### 2.4.2. Quality Control of Methylation Data

Filtered data were normalized using the Beta Mixture Quantile (BMIQ) method [35]. With the Human MethylationEPIC BeadChip (850K), a total of 866,297 CpG sites were extracted for the raw data, and 609 CpG sites (0.07%) which had detection *p*-value ≥ 0.05 across more than 25% of all samples were excluded from analysis. Thus, 865,688 CpG sites were left for analysis. We also filtered CpG sites according to the following exclusion criteria: (a) SNP-associated CpG sites defined as 0 or 1 base pair near SNP loci or minor allele frequency (MAF) > 5% (213,660 CpG sites); (b) CpG sites that corresponded to the X or Y chromosome (19,627 CpG sites); (c) CpG sites corresponding to non-CpG loci (3627 CpG sites).; (d) cross-reactive CpG sites (42,558 CpG sites). We were finally left with 256,866 CpG sites which overlapped with the available epigenome data of 6-year-old children for further analysis. We also excluded multimodal CpG sites if they appeared in statistically significant CpG sites, which were identified using the dip test statistic for multimodality, which was calculated using the R package *diptest* module [36].

### 2.5. Measurement of Intelligence Quotient in Children

The IQ of the 6-year-old children was measured using the Korean Educational Developmental Institute’s Wechsler Intelligence Scale for Children [37]. Higher scores indicated higher IQ. Two subsets were measured: verbal IQ, based on the sum of the test results for vocabulary and arithmetic intelligence, and performance IQ, based on the sum of the tests for picture arrangement and block design [38].

### 2.6. Measurement of Other Exposure Variables

We collected demographic information on the children and their mothers by means of interviews using structured questionnaires. Covariates were selected based on a literature review [27,30]. The covariates used to analyze the association between greenness and DNA methylation were mother’s age at pregnancy (years), mother’s educational level (middle school graduate, high school graduate, college graduate, or graduate school attendance), children’s exposure to environmental tobacco smoke (ETS) at age 2 (yes or no), children’s sex (male or female), children’s age at follow-up (months, continuous variable), children’s body mass index (BMI) (kg/m^2^, continuous variable), and cell type fractions (continuous variables). The cell type fractions in blood samples were calculated using the R package *minfy* module [39]. To estimate the percentage of CD8 + T cells, CD4 + T cells, natural killer cells, B cells, monocytes, and neutrophils, adults’ leukocyte reference dataset was used [40]. We also used the covariates for analyzing the association between DNA methylation and children’s IQ at age 6, including children’s age at follow-up, children’s BMI, maternal age during pregnancy, maternal education level, exposure to ETS at age 2, maternal IQ, and children’s sex. The short form of the Korean Wechsler Adult Intelligence Scale was used to assess maternal IQ at the time of their children’s follow-up visit at the age of 6 [41].

### 2.7. Statistical Analysis

We compared the demographic and clinical characteristics of the sub-study population to the population of the EDC study that was not included in our study using the Student’s *t*-test (for continuous variables) or chi-square test (for categorical variables) (Table 1). We used batch-effect-adjusted DNA methylation data obtained using the R package *ComBat* module to adjust different distributions according to chips and positions from the array data [42]. This process uses an empirical Bayes method to adjust batch effects in small sample sizes. We performed multivariable linear regression to determine the relationship between exposure to greenness at age 2 and cognitive-ability-related DNA methylation at age 2, adjusting for monthly age at follow-up, BMI at age 2, maternal education level, cell type fractions (CD8 + T cells, CD4 + T cells, natural killer cells, B cells, monocytes, and neutrophils), ETS, maternal age at delivery, and child’s sex. Using the CpG sites significantly associated with exposure to greenness at age 2, we tested the association of DNA methylation levels at these CpG sites with total, verbal, and performance IQ scores at age 6, adjusting for maternal age during pregnancy, exposure to ETS at age 2, maternal IQ, and children’s sex using the selection criteria to select the best model by comparing the Akaike Information Criterion (AIC) [43]. Using the CpG sites significantly associated with exposure to greenness at age 2, we tested the association of DNA methylation levels at these CpG sites with total, verbal, and performance IQ scores at age 6 in multiple linear regression models, adjusting for maternal age during pregnancy, exposure to ETS at age 2, maternal IQ, and children’s sex using the selection criteria to select the best model by comparing the AIC. To account for multiple testing, we used a Bonferroni-corrected *p*-value ≤ 0.05 for statistical significance. Pathway enrichment analysis was performed using the *ReactomePA* R package [44]. Enrichment analysis of functional terms revealed the Reactome pathway enriched in the genes identified as significant from their association between greenness exposure and cognitive-ability-related CpG sites (Bonferroni-corrected *p* ≤ 0.05). All statistical analyses were performed using SAS version 9.4 (SAS Institute Inc., Cary, NC, USA) and R software version 3.6.0 (R Foundation for Statistical Computing, Vienna, Austria).

## 3. Results

### 3.1. Participant Characteristics

Table 1 presents the participant characteristics. The mean maternal age at delivery was 31.10 years (standard deviation (SD): 3.79 years). The mean age and BMI of children were 23.32 months (SD: 0.77 months) and 16.57 kg/m^2^ (SD: 1.20 kg/m^2^), respectively. The mean maternal IQ was 117.8 (SD: 11.5). The percentages of mothers who received less than a high school education and more than a graduate school education were 15.25% and 13.56%, respectively. A total of 23.73% of the participants were in a group with positive exposure to ETS during pregnancy. There were similar numbers of girls and boys in the study (30 and 29, respectively). The percentage of greenness exposure at age 2 within 100–2000 m was ranged from 17.67% to 25.29%. The mean total, verbal, and performance IQ scores at age 6 were 107.40 (SD: 13.70), 21.08 (SD: 5.03), and 23.47 (SD: 5.10), respectively. In addition, we found that the characteristics of our study subcohort were not significantly different from those of the participants in the entire EDC cohort, except for exposure to greenness at age 2 in 1000 m buffer of residential address.

### 3.2. Systematic Literature Review

We found a total of 896 studies (445 studies in PubMed and 451 studies in EMBASE) after applying the keywords search strategy described in Appendix A. Five studies were included in the bibliographic search. After excluding duplicated studies (*n* = 64), 97 studies were included for screening by title, and 735 studies were excluded because they were studies of cognitive aging or cognitive disease or were not primary investigation. We further excluded irrelevant articles such as invalid study designs or cognitive outcomes, such as mathematics, school performance, or memory, finally leaving a total of 19 articles (Figure 1).

A total of 400 CpG sites were selected from 6 EWAS [26,45,46,47,48,49]. Additionally, a total of 31,981 CpG sites were selected, which were annotated to 835 genes reported from 13 GWAS after excluding duplicate genes [50,51,52,53,54,55,56,57,58,59,60,61,62]. As a result, 8743 CpG sites were finally selected (Appendix A).

### 3.3. Association between Greenness Exposure and DNA Methylation

A total of 209 CpG sites from the EWAS and 8,534 CpG sites from the GWAS were analyzed in our study. We found that 25 cognitive-ability-related CpG sites were significantly associated with greenness exposure at age 2 (8 CpG sites from EWAS and 17 CpG sites from GWAS) (Table 2) in total greenness in buffers of 100–2000 m, natural greenness in buffers of 1000–2000 m, and built greenness in buffers of 1000 m and 1500 m, with a significance criterion for Bonferroni-corrected *p*-values < 0.05 (Table 2).

### 3.4. Pathway Enrichment Analysis

We investigated potential biological functions by performing pathway enrichment analysis with the cutoff *p*-value set to 0.1. We found the top 20 pathways, including transmission across chemical synapses, opioid signaling, and neuronal systems pathway (Figure 2A). Notably, a single pathway of neurotransmitter clearance was only significantly enriched for the *SLC6A3* and *SLC6A4* genes at the selected cutoff (*p*-value was 0.05) (Appendix A). *SLC6A3* and *SLC6A4* genes were significantly related to greenness exposure, of which *SLC6A3* also showed significant associations with IQ in this study. Figure 2B shows linkages between the genes and biological functions as a network. In addition to the neurotransmitter clearance pathway, *SLC6A4* and *SLC6A3* were non-significantly linked via transmission across chemical synapses (adjusted *p*-value: 0.12, respectively; Appendix A).

### 3.5. Association between DNA Methylation and Children’s Intelligence Quotient

The association of the methylation levels of the 25 CpG sites at age 2 and total, verbal, and performance IQ scores at age 6 are shown in Appendix A. Notably, one interquartile range (IQR) increase of the methylation level at cg26269038 was significantly associated with an increased performance IQ score at age 6 (2.89-point increase in IQ was 2.89 (95% CI: 1.27, 4.51) in the adjusted models after Bonferroni correction (Table 3). However, there was no significant association between total IQ or verbal IQ and the level of DNA methylation (Table 3). In sensitive analysis, we analyzed these associations with additional covariates such as breastfeeding pattern and mother’s previous smoking status in Appendix A. The result was not different from the main result in Table 3. We plotted the least-squares means of the methylation level at cg26269038 by exposure to greenness as the quartile group. The percentages of greenness in the 100 m buffer of residential address for each participant were divided into quartiles and were then performed to determine whether individuals in the three higher quartiles differed significantly from those in the lowest quartile. The highest quartile of DNA methylation level at cg26269038 was significantly different from the lowest quartile (Figure 3).

## 4. Discussion

We found that the methylation levels at 25 cognitive ability-related CpG sites at age 2 were significantly associated with greenness exposure during early childhood and that the methylation level at cg26269038 at age 2 (*SLC6A3*) was significantly associated with the performance IQ score at age 6.

Epigenetic markers, such as DNA methylation, are dynamically reprogrammed during gametogenesis and early embryo preimplantation [63,64]. Experimental evidence suggests that the epigenome of mammalian embryonic cells is more susceptible to environmental stimulation than other differentiated cells [18,65] because of the abundance of de novo DNA methyltransferases in these rapidly dividing pluripotent cells [63,64]. The most significant DNA methylation change at cg26269038 is located in the body, intron between the third and fourth exon, of the gene solute carrier family 6, the member 3 (*SLC6A3*) on the chromosome 5. The gene encodes a dopamine transporter (*DAT*), which is a member of the sodium- and chloride-dependent neurotransmitter transporter family, and provides rapid clearance of dopamine [66,67], which mediates the reuptake of dopamine from the synaptic cleft [68]. Cómbita et al. (2017) determined whether *SLC6A3/DAT1* gene contributed to individual differences in children’s self-regulation skills [69]. They evaluated self-regulation skills and cognitive tasks such as conflict processing, inhibitory control, and intelligence assessments in 127 children at ages 4 and 6 in Spain. They found that the presence of the 10 alleles of the *SLC6A3* gene was related to a declining function of the dopaminergic transmission system, which was associated with poorer performance in self-regulation. Dopaminergic neurotransmission related to the *SLC6A3* and *DRD2* genes is reportedly associated with cognitive capacities, such as IQ, in previous studies [70,71,72]. Our results show that children with greater exposure to greenness had lower DNA methylation levels of the *SLC6A3* gene. This region might be linked to greenness exposure and neurological development in children. However, further studies are needed to understand how these cognitive-ability-related CpG sites are linked to greenness exposure. In line with previous findings, we found that greenness exposure in early childhood is a modifiable factor related to DNA methylation change, which was found to be associated with cognitive ability in a previous study.

In our study, several other genes also significantly associated with greenness exposure, including *PDE4D, PLCL1, GNG12*, and *SLC6A4*, were also linked to neurotransmitter clearance in the pathway-enrichment analysis results. Signaling in the central nervous system (CNS) is terminated by the clearance of neurotransmitters from the synapse via high-affinity transporter molecules in the presynaptic membrane [73]. Accumulated evidence has shown that exposure to greenness has a positive effect on health by reducing oxidative stress [74,75,76,77]. Additionally, oxidative-stress-induced damage to the brain is likely to negatively affect normal CNS functions [78]. As these neurodegenerative disorders are related to increased environmental stressors, toxins, and oxidative stress in adults [79], brain development in children may also be linked to oxidative stress, which is reduced by greenness exposure. Similarly, oxidative stress is widely related to brain development. Recently, greenness exposure was significantly associated with reduced oxidative stress in Italian children [75]. We suggested that exposure to greenness, which was a pathway to reducing oxidative stress, may be involved in neurodevelopment.

A CpG site at cg05016953 (*SLC6A4*) as a serotonin transporter, which was significantly associated with greenness exposure in our study, was reported to be regulated by a *5HTTLPR* functional polymorphism, which was significantly associated with IQ scores in a previous study [80]. However, our results showed no significant association between DNA methylation changes at cg05016953 (*SLC6A4*) at age 2 and children’s IQ at age 6. As there is a wide distribution of the *5HTTLPR* genotype by race and ethnicity [81,82,83], further studies should be conducted among Asian children.

The effect sizes of the association between residential greenness and DNA methylation were within 1~3%. In an Australian study that investigated the association between greenness and epigenome-wide DNA methylation, the coefficients ranged from−0.36% to 1.73% [27]. Epidemiological studies concerning the effects of environmental exposures typically show small effect sizes. For example, the differences in DNA methylation reported between exposed vs. unexposed groups are generally on the scale of 2–10%, and in some cases, even smaller differences have been observed [84]. It has been reported that for every 1% change in methylation at the differentially methylated region at *IGF2*, a halving or doubling of *IGF2* transcription was observed [85]. Although such a few percent change in DNA methylation appears as a small effect size, it is only so in the perspective of the population of cells. At a single-cell level, a CpG site is either methylated (100%), hemi-methylated (50%), or non-methylated (0%), and such a difference could have substantial effects on cell functions, including gene regulation [84].

Exposure to different types of greenness has been shown to have different effects on children’s health. A previous study found a strong association between exposure to built greenness, but not natural greenness, and children’s IQ at age 6 [30]. However, DNA methylation changes were more significantly associated with natural greenness at age 2. There have been no previous studies of the association between the type of greenness and DNA methylation change, and further studies are needed to investigate the effect of exposure to various types of greenness and intelligence in children. The DNA methylation changes and greenness exposure still need to be analyzed in the entire EDC cohort rather than the sub-study [30].

Our study had several strengths. First, to our knowledge, this is the first epigenetic study of the association between greenness exposure and cognitive-ability-related DNA methylation changes in children. Second, we estimated the association between greenness exposure related to DNA methylation at age 2 and children’s IQ at age 6 in a prospective cohort study, which might provide a clue to explain the causal role of greenness in neurodevelopment in children. Third, we estimated the proportions of greenness in buffers of various sizes and exposure to various types of greenness. There is currently insufficient evidence to determine which type of greenness exposure and which buffer size have the greatest impact on mental health in children, and further studies should be performed using buffers of various sizes and various types of greenness.

However, our study has some limitations. First, we only had DNA methylation data of 59 children, not the entire EDC cohort, even though we found no significant differences in the characteristics of the children in the sub-cohort and the full EDC cohort; thus, our findings need to be further evaluated in a larger cohort. Second, we measured the surrounding greenness captured during a single period using satellite-derived land-cover maps without considering the period of greenness assessment. Third, we focused on residential surrounding greenness based on the residential addresses of our participants, which did not reflect their exposure level to greenness at other places, which may have caused exposure misclassification. Fourth, we were unable to evaluate participants’ access to greenness due to the lack of information, so further study is needed to consider how children’s exposure to greenness is related to their accessibility to greenness.

## 5. Conclusions

Surrounding greenness exposure at age 2 was associated with DNA methylation changes, and further associated with cognitive abilities. Further studies are warranted to clarify the epigenetic pathways linking greenness exposure and neurocognitive functions in children.

## Figures and Tables

**Figure 1 ijerph-18-07429-f001:**
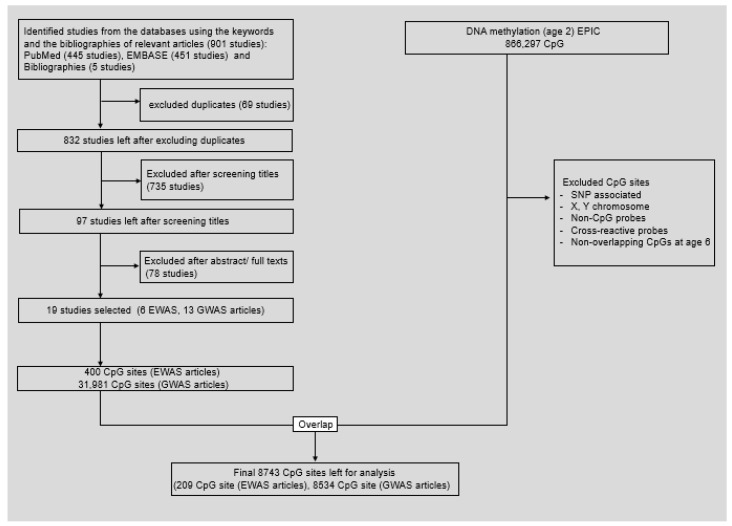
Workflow for model building for selecting cognitive abilities based on a systematic literature review.

**Figure 2 ijerph-18-07429-f002:**
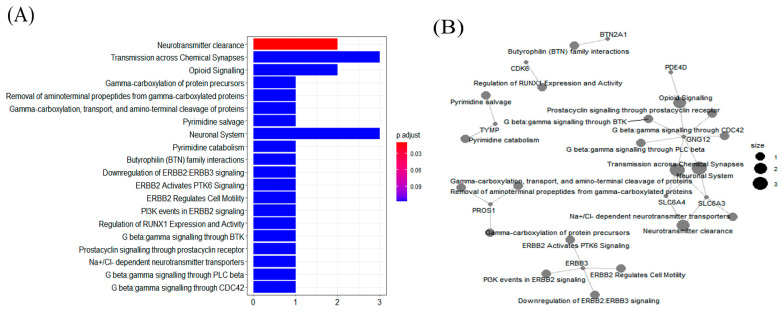
Reactome pathway enrichment analysis of the greenness-associated genes, which shows the top 20 pathways. (**A**) The enrichment scores in the Reactome pathway analysis of the greenness exposure-related genes. (**B**) The network of the most enriched pathways of the greenness exposure-related genes. Larger nodes represent higher enrichment scores that represent different enrichment modules.

**Figure 3 ijerph-18-07429-f003:**
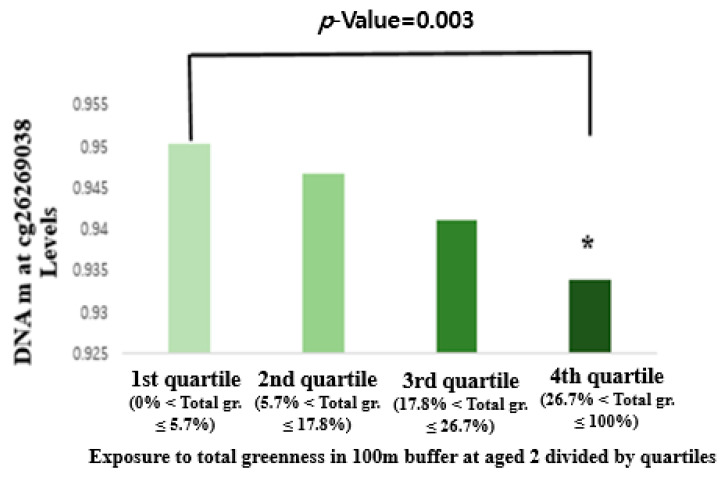
The LSMEANS of DNA m at cg26269038 levels in four quartiles of exposure to greenness among children. * Represents significant differences between a quartile and the lowest quartile, with *p*-values < 0.05 considered statistically significant.

**Table 1 ijerph-18-07429-t001:** Characteristics of participants at age 2 in sub-study compared to total EDC population.

Variables		Study Population (*n* = 59)	EDC Population Excluded from the Study (*n* = 366)	*p*-Value
		*n* (%) or Mean ± SD	*n* (%) or Mean ± SD	
Maternal age at pregnancy (years)		31.10 ± 3.79	31.68 ± 3.60	0.256
Children’s age (months)		23.32 ± 0.77	23.31 ± 0.76	0.922
Children’s BMI at age 2		16.57 ± 1.20	16.48 ± 1.44	0.666
Maternal IQ		117.8 ± 11.5	115.8 ± 11.1	0.248
Maternal education level	High school graduate	9 (15.25)	70 (19.13)	0.669
	College graduate	42 (71.19)	257 (70.22)	
	Graduate school	8 (13.56)	39 (10.66)	
Prenatal exposure to ETS	Yes	14 (23.73)	89 (24.32)	0.922
	No	45 (76.27)	277 (75.68)	
Children’s sex	Girl	30 (50.85)	172 (46.99)	0.582
	Boy	29 (49.15)	194 (53.01)	
Percentage of total greenness at the home address at age 2	100 m	17.67 ± 12.8	19.82 ± 14.0	0.276
	500 m	18.71 ± 11.1	21.36 ± 13.9	0.108
	1000 m	19.95 ± 11.2	23.61 ± 13.6	0.028
	1500 m	23.63 ± 12.8	24.77 ± 12.9	0.529
	2000 m	25.29 ± 13.4	26.32 ± 12.7	0.569
IQ at age 6	Total IQ	107.4 ± 13.7	110.6 ± 12.5	0.088
	Verbal IQ	21.08 ± 5.03	20.81 ± 7.06	0.732
	Performance IQ	23.47 ± 5.10	22.91 ± 7.26	0.487

Abbreviations: EDC, Environment and the Development of Children; SD, standard deviation; ETS, environmental tobacco smoke; BMI, body mass index; IQ, intelligence quotient.

**Table 2 ijerph-18-07429-t002:** The significant relationship between greenness exposure and selected DNAm at age 2 ^†^.

Origin Study	Greenness Type	Buffer	CpG	Gene	Difference (95% CI) ^§^	*p*-Value
EWAS Study	Total	100 m	cg13092901	*TYMP*	0.021(0.012, 0.029)	2.0 × 10^−5^
		500 m	cg04789403	*NA*	0.031(0.015, 0.047)	1.1 × 10^−4^
			cg07266431	*CDK6*	0.028(0.015, 0.040)	1.9 × 10^−4^
			cg13599020	*SAMD3*	0.026(0.014, 0.039)	1.9 × 10^−4^
			cg27492942	*CISD3*	0.029(0.013, 0.045)	1.7 × 10^−4^
		1000 m	cg00252813	*GAPDH*	0.013(0.006, 0.020)	6.1 × 10^−5^
	Natural	1000 m	cg00252813	*GAPDH*	0.013(0.007, 0.019)	5.8 × 10^−5^
			cg04789403	*NA*	0.020(0.010, 0.030)	7.7 × 10^−5^
	Built	1000 m	cg16594502	*NA*	0.015(0.008, 0.022)	9.4 × 10^−5^
		1500 m	cg25189904	*GNG12*	0.028(0.013, 0.044)	2.3 × 10^−4^
GWAS Study	Total	100 m	cg26269038	*SLC6A3*	−0.011(−0.015, −0.007)	3.2 × 10^−8^
			cg14464361	*AGAP1*	−0.023(−0.032, −0.016)	2.2 × 10^−6^
			cg21175642	*CELSR3*	0.013(0.007, 0.016)	3.4 × 10^−6^
		1000 m	cg23651585	*AUTS2*	−0.039(−0.056, −0.024)	9.9 × 10^−7^
			cg27636559	*EFTUD1*	0.007(0.004, 0.009)	1.2 × 10^−6^
			cg27609819	*PLCL1*	−0.027(−0.038, −0.015)	2.3 × 10^−6^
			cg16296679	*WBP2NL*	0.015(0.009, 0.022)	2.9 × 10^−6^
		1500 m	cg17146029	*AUTS2*	0.010(0.007, 0.013)	1.0 × 10^−7^
			cg00809988	*ELAVL2*	−0.006(−0.009, −0.003)	1.5 × 10^−7^
		2000 m	cg17146029	*AUTS2*	0.009(0.006, 0.012)	3.9 × 10^−8^
			cg00809988	*ELAVL2*	−0.004(−0.008, −0.002)	3.2 × 10^−7^
			cg03367519	*PDE4D*	−0.005(−0.008, −0.002)	3.3 × 10^−6^
	Natural	1000 m	cg27609819	*PLCL1*	−0.029(0.039, −0.020)	3.7 × 10^−8^
			cg23651585	*AUTS2*	−0.043(−0.059, −0.027)	7.4 × 10^−8^
			cg27636559	*EFTUD1*	0.007(0.005, 0.009)	2.2 × 10^−7^
		1500 m	cg23651585	*AUTS2*	−0.041(−0.055, −0.025)	2.6 × 10^−7^
			cg23159678	*NOVA1*	0.009(0.004, 0.014)	1.9 × 10^−6^
			cg05016953	*SLC6A4*	−0.004(−0.006, −0.001)	2.2 × 10^−6^
			cg27609819	*PLCL1*	−0.025(−0.036, −0.016)	2.2 × 10^−6^
			cg03367519	*PDE4D*	−0.005(−0.007, −0.002)	2.9 × 10^−6^
			cg00809988	*ELAVL2*	−0.005(−0.007, −0.002)	5.3 × 10^−6^
		2000 m	cg17146029	*AUTS2*	0.010(0.006, 0.012)	1.8 × 10^−6^
			cg23651585	*AUTS2*	−0.046(−0.064, −0.026)	1.9 × 10^−6^
			cg11176256	*BAIAP2*	0.016(0.010, 0.023)	3.5 × 10^−6^
			cg05897638	*PROS1*	−0.007(−0.010, −0.003)	5.1 × 10^−6^
			cg00809988	*ELAVL2*	−0.005(−0.009, −0.002)	5.5 × 10^−6^
			cg12414502	*BTN2A1*	0.010(0.006, 0.012)	5.6 × 10^−6^
	Built	1500 m	cg19258882	*ERBB3*	0.024(0.015, 0.032)	4.6 × 10^−6^
			cg18311871	*PTPRN2*	0.081(0.047, 0.115)	3.2 × 10^−6^

Abbreviations: CpG site location based on Illumina annotation, derived from the University of California, Santa Cruz (UCSC), adjusted for children’s age, children’s BMI, maternal education level, cell type fractions (CD8 + T cells, CD4 + T cells, natural killer cells, B cells, monocytes, and neutrophils), environmental tobacco smoke, maternal age, and children’s sex. The list was significantly associated as per Bonferroni correction (*p* < 0.05). ^†^ Analyzed using a linear regression model. § Change in DNA methylation level by an increase of 1 interquartile range of greenness percentage within each buffer.

**Table 3 ijerph-18-07429-t003:** Association between selected CpG sites and children’s Performance IQ (*n* = 59) †.

IQ	CpG Sites	Chr	Gene	Gene Group	Difference (95% CI) §	*p*-Value *
Total IQ	cg26269038	5	*SLC6A3*	Body	3.68(−0.90, 8.26)	0.115
Verbal IQ					−0.66(−2.47, 1.15)	0.475
**Performance IQ**					**2.89(1.27, 4.51)**	**0.001**

* Bold was significant association using Bonferroni correction *p* < 0.002. † Analyzed using a linear regression model. § Difference (95% CI) was calculated by 1 interquartile range change in DNA methylation level at each CpG site. Adjusted for children’s sex, maternal age during pregnancy, exposure to ETS, and maternal IQ.

## Data Availability

Not applicable.

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
