# Peer review of "Children’s Greenness Exposure and IQ-Associated DNA Methylation: A Prospective Cohort Study"

_ijerph, 2021, doi:10.3390/ijerph18147429_

Round 1
Reviewer 1 Report
In this study, Lee et al. examine the associations between estimated greenness exposure and IQ with childhood measures of genome-wide DNA methylation. To strengthen the biological underpinnings of their analysis and limit the scope, they have limited their analysis to only CpG sites that are expected to show some association with cognition.
The manuscript uses a data-rich human cohort to test their hypotheses, and largely presents their data in an appropriate manner, but the study is held back by some serious methodological concerns, including high levels of CpG probe failure, small sample size, and potential model overfitting.
Section 2.1: Of the n=59 sub-study participants, how many were male and how many were female?
Section 2.2: Did the authors consider using gene ontology or KEGG pathways to identify genes related to cognitive function/development?
Section 2.4.1: “then converted to bisulfite” should read “bisulfite converted.” The DNA is not being converted to bisulfite; sodium bisulfite is being used to deaminate unmethylated thymines in the DNA.
Section 2.4.2: “CpG sites with a detection p-value ≥ 0.05, which comprised more than 25% of the samples, were excluded, which were likely to be noise signals.” The note that more than 25% of samples had CpG sites with detection p-value > 0.05 is both confusing and concerning. What is meant by “comprised more than 25% of the samples?” Is “samples” referring to CpG sites? Or to the DNA samples? Either way, this is an extremely high level of detection p-values > 0.05. Normally, studies will exclude any samples where greater than 5-10% of CpG sites have detection p-value >0.05 (see example for 5% cutoff for samples using pfilter function in wateRmelon: https://www.ncbi.nlm.nih.gov/pmc/articles/PMC6429761/), as it can suggest that BeadChip hybridization was not very successful. This is a major concern with this dataset. How many CpG sites had detection p-value >0.05 per sample? If it’s truly ~25%, then that suggests a quarter of the array is nothing but noise. This is not typical for an Illumina BeadChip array, and suggests serious degradation of DNA prior to hybridization. It also makes the data quite difficult to interpret, as the results may be confounded by this degradation, which may not be equal across the genome. The authors should either make this potential confounding clear in their discussion or limit their analyses to only samples where 5-10% (or less) of CpGs had detection p-values >0.05.
Section 2.4.2: The authors do not say whether they removed cross-reactive probes from their analysis, which is an important step in all EWAS studies using the Illumina EPIC array. See documentation of EPIC array cross-reactive probes here: https://pubmed.ncbi.nlm.nih.gov/27924034/. These probes must be removed, as they can cause spurious results.
Section 2.6: “We also used the covariates for analyzing the association between DNA methylation and children’s IQ at age 6, including children’s age at follow-up, children’s BMI, maternal age during pregnancy, maternal education level, exposure to ETS at age 2, maternal IQ, and children’s sex.” This is a huge number of covariates for only n=59 samples. Were all of these included in each model? If so, I would have concerns about overfitting. Generally, you want at least 10-15 observations per term in a regression model. This is not the case for the 8 terms listed here. The authors should consider rerunning their analyses with a maximum of 5 or 6 terms, using selection criteria (e.g. AIC values) to select the best model.
Section 2.7: After filtering and processing steps, how many CpG sites were analyzed using the described modeling approach?
Section 2.7: “Using the CpG sites significantly associated with exposure to greenness at age 2, we tested the association of these sites with total, verbal, and performance IQ scores at age 6, adjusting for children’s age at follow-up, children’s BMI, maternal age during pregnancy, maternal education level, exposure to ETS at age 2, maternal IQ, children’s sex, and cell type fractions.” I assume you tested the association between the beta values for DNA methylation at these sites with your predictors? If so, please make this clear in the text.
Table 2: “Change in DNA methylation by 1 interquartile range increases the proportions of greenness within the given radii from a residential address.” This statement is unclear. What does “by interquartile range increases the proportions of greenness” mean? Is this the change in DNA methylation by an increase of 1 in the interquartile range of greenness? And futher, are these percentages of DNA methylation or raw beta values? Please clarify by improving language here.
Table 2: What do the authors mean by a “robustly detected gene”? This is not explained anywhere in the text.
Table 2: A lot of the significant sites appear to show 0.000 as the change in DNA methylation by greenness. This suggests false positives could be occuring, possibly as a result of overfitting. Do the authors have an alternate explanation for these results?
Section 3.4: In the methods, the authors described using a Bonferroni-corrected P ≤ 0.05 cutoff for detection of significant pathways. It appears only the top pathway in Figure 3 reaches this cutoff, so including the other non-significant pathways in Figure 3.4 is rather misleading. Please either remove these pathways from the figure or clarify in the text that only a single pathway -- neurotransmitter clearance – was significant at the selected cutoff.
Figure 2: The authors do not provide any description of Figure 3B in the text. Please provide language to assist readers with interpretation of this plot.
Section 3.5: The authors do not provide clear language describing the fact that they re-analyzed the 25 CpG sites that were significant in the greenness models in new models using IQ. As such, the choice feels a bit arbitrary, especially since almost none of the CpG sites show a significant association with IQ. This is a lot of negative data for such a large table; consider better describing the results in the text and limiting data table to only sites that show significance in the follow-up, IQ model.
Figure 3: The x-axis label on this chart is unclear. What does “Exposure to total greenness in 100 m buffer at aged 2” referring to? Isn’t this a graph of quartiles? If so, please clarify the axis label to better explain the units.
Discussion: “Recently, a review article reported that disruption in the clearance of neurotransmitters and increased amyloid β and tau from astrocytes appeared to be involved in neurotoxicity in Alzheimer’s disease patients”. The authors include this information, but then do little to build upon it. How does this related to the results in the discussion?
Discussion: The discussion jumps between topics –DNA methyltransferases, greenness exposure, Parkinson’s disease, oxidative stress -- with little apparent follow-up or logical flow. The information presented is not wrong, but re-organization of the ideas into a clearer story is strongly suggested. For example, each paragraph could touch on one of these topics in depth, examining how the new results might play into what’s already known.
Discussion: The authors do not provide a discussion of whether the effect sizes they see (e.g. ~1-2% change in DNAm at SLC6A3 gene) are expected to be functionally relevant. Would this level of DNA methylation change actually impact gene regulation?
Author Response
Thank you for providing us another opportunity to improve the quality of our study. We have further revised the manuscript according to the reviewers’ valuable and helpful comments, and have provided our point-by-point responses to each comment.
Point 1: In this study, Lee et al. examine the associations between estimated greenness exposure and IQ with childhood measures of genome-wide DNA methylation. To strengthen the biological underpinnings of their analysis and limit the scope, they have limited their analysis to only CpG sites that are expected to show some association with cognition.
The manuscript uses a data-rich human cohort to test their hypotheses, and largely presents their data in an appropriate manner, but the study is held back by some serious methodological concerns, including high levels of CpG probe failure, small sample size, and potential model overfitting.
Section 2.1: Of the n=59 sub-study participants, how many were male and how many were female?
Response 1: Thank you for your comment. We reported that boy was 29, and girl was 30 in Table 1.
Point 2: Section 2.2: Did the authors consider using gene ontology or KEGG pathways to identify genes related to cognitive function/development?
Response 2: Thank you for your comment. Our selected genes for cognitive abilities were based on literature review from EWAS or GWAS studies. However, we checked whether these studies performed functional enrichment analysis. We added the information in Supplementary Table1. In addition, we included two bibliographies, which selected IQ-related genes using functional enrichment analyses. We also added the information in our manuscript as follows.
|
Before revision |
|
2.2. Systematic review of literature and selection of candidate cytosine-guanine dinucleotide sites As we were specifically interested in the question of whether DNA methylation mediates the effects of exposure to greenness on children’s IQ, we targeted CpG sites that were more likely to be involved in cognitive ability instead of scanning the whole epigenome. For the selection of previous EWAS or GWAS on association with cognitive abilities, we searched PUBMED and EMBASE on April 1, 2021, using keywords (“epigenome-wide association study” or “genome-wide association study”) and (“intelligence” or “cognitive ability” or “cognitive development”) from titles or abstracts. The selection criteria were EWAS or GWAS regarding cognitive ability in healthy children or adults. From previous EWAS or GWAS that investigated the association between DNA methylation and cognitive ability in healthy children or adults, we identified CpG sites associated with cognitive ability-related parameters (Figure 1). In the GWAS, single nucleotide polymorphisms (SNPs) associated with cognitive ability were identified, and then the genes annotated to these SNPs were identified. The CpG sites associated with these genes were pooled using the Database for Annotation, Visualization, and Integrated Discovery (DAVID, http://david.abcc.ncifcrf.gov/home.jsp). |
|
After revision |
|
2.2. Systematic review of literature and selection of candidate cytosine-guanine dinucleotide sites As we were specifically interested in the question of whether DNA methylation mediates the effects of exposure to greenness on children’s IQ, we targeted CpG sites that were more likely to be involved in cognitive ability instead of scanning the whole epigenome. For the selection of previous EWAS or GWAS on association with cognitive abilities, we searched PUBMED and EMBASE on April 1, 2021, using keywords (“epigenome-wide association study” or “genome-wide association study”) and (“intelligence” or “cognitive ability” or “cognitive development”) from titles or abstracts. The selection criteria were EWAS or GWAS regarding cognitive ability in healthy children or adults. From previous EWAS or GWAS that investigated the association between DNA methylation and cognitive ability in healthy children or adults, we identified CpG sites associated with cognitive ability-related parameters (Figure 1). In the GWAS, single nucleotide polymorphisms (SNPs) associated with cognitive ability were identified, and then the genes annotated to these SNPs were identified. The CpG sites associated with these genes were pooled using the Database for Annotation, Visualization, and Integrated Discovery (DAVID, http://david.abcc.ncifcrf.gov/home.jsp). We added IQ-related genes from bibliographies articles, which were selected by enrichment analyses. |
Point 3: Section 2.4.1: “then converted to bisulfite” should read “bisulfite converted.” The DNA is not being converted to bisulfite; sodium bisulfite is being used to deaminate unmethylated thymines in the DNA.
Response 3: you for the advice. We apologize for the confusion we have made. We meant DNA is bisulfite converted. We changed the sentence as the following.
|
Before revision |
|
The diluted gDNA samples (minimum 500 ng) were then converted to bisulfite using the Zymo EZ DNA methylation kit (Zymo Research, Irvine, CA, USA) |
|
After revision |
|
The gDNA samples (minimum 500 ng) were diluted, then bisulfite-converted using the Zymo EZ DNA methylation kit (Zymo Research, Irvine, CA, USA) |
Point 4: Section 2.4.2: “CpG sites with a detection p-value ≥ 0.05, which comprised more than 25% of the samples, were excluded, which were likely to be noise signals.” The note that more than 25% of samples had CpG sites with detection p-value > 0.05 is both confusing and concerning. What is meant by “comprised more than 25% of the samples?” Is “samples” referring to CpG sites? Or to the DNA samples? Either way, this is an extremely high level of detection p-values > 0.05. Normally, studies will exclude any samples where greater than 5-10% of CpG sites have detection p-value >0.05 (see example for 5% cutoff for samples using pfilter function in wateRmelon: https://www.ncbi.nlm.nih.gov/pmc/articles/PMC6429761/), as it can suggest that BeadChip hybridization was not very successful. This is a major concern with this dataset. How many CpG sites had detection p-value >0.05 per sample? If it’s truly ~25%, then that suggests a quarter of the array is nothing but noise. This is not typical for an Illumina BeadChip array, and suggests serious degradation of DNA prior to hybridization. It also makes the data quite difficult to interpret, as the results may be confounded by this degradation, which may not be equal across the genome. The authors should either make this potential confounding clear in their discussion or limit their analyses to only samples where 5-10% (or less) of CpGs had detection p-values >0.05.
Response 4: We apologize for the confusion regarding the expression. We intended “CpG sites with a detection p-value ≥0.05 in more than 25% of the sample were excluded”, not that 25% of samples had CpG sites with detection p-value > 0.05. In our analysis, a total of 866,297 CpG sites were extracted for the raw data. Out of this CpG sites, 609 CpG sites (0.07%) had detection p-value ≥0.05 across more than 25% of all samples, thus excluded from analysis. Therefore, 865688 CpG sites were left for analysis. We changed the sentence as the following.
|
Before revision |
|
Filtered data were normalized using the Beta Mixture Quantile (BMIQ) method [35]. With the Human MethylationEPIC BeadChip (850K), a total of 865,688 CpG sites among the 866,297 CpG sites remained after excluding CpG sites with "not-available" (NA) val-ues for at least one sample. CpG sites with a detection p-value ≥ 0.05, which comprised in more than 25% of the samples, were excluded, which were likely to be noise signals. |
|
After revision |
|
Filtered data were normalized using the Beta Mixture Quantile (BMIQ) method [35]. With the Human MethylationEPIC BeadChip (850K), a total of 866,297 CpG sites were extracted for the raw data, and 609 CpG sites (0.07%) which had detection p-value ≥0.05 across more than 25% of all samples were excluded from analysis. Thus, 865688 CpG sites were left for analysis. |
Point 5: Section 2.4.2: The authors do not say whether they removed cross-reactive probes from their analysis, which is an important step in all EWAS studies using the Illumina EPIC array. See documentation of EPIC array cross-reactive probes here: https://pubmed.ncbi.nlm.nih.gov/27924034/. These probes must be removed, as they can cause spurious results.
Response 5: Thank you for your advice. We have checked the cross-reactive probes according to McCartney et al, 2016. We excluded these cross-reactive CpG sites from our analysis, and we obtained the consistent results. We added this procedure in the manuscript.
McCartney, Daniel L., et al. "Identification of polymorphic and off-target probe binding sites on the Illumina Infinium MethylationEPIC BeadChip." Genomics data 9 (2016): 22-24. https://doi.org/10.1016/j.gdata.2016.05.012
|
Before revision |
|
Method We also filtered CpG sites according to the following exclusion criteria: a) SNP-associated CpG sites defined as 0 or 1 base pair near SNP loci or minor allele frequency (MAF) > 5%; b) CpG sites that corresponded to the X or Y chromosome; c) CpG sites corresponding to non-CpG loci.
|
|
After revision |
|
Method We also filtered CpG sites according to the following exclusion criteria: a) SNP-associated CpG sites defined as 0 or 1 base pair near SNP loci or minor allele frequency (MAF) > 5% (213,660 CpG sites); b) CpG sites that corresponded to the X or Y chromosome (19,627 CpG sites); c) CpG sites corresponding to non-CpG loci (3,627 CpG sites).; d) cross-reactive CpG sites (42,558 CpG sites). We were finally leaving 256,866 CpG sites to overlap the available data of 6 years-old for further analysis. |
Point 6: Section 2.6: “We also used the covariates for analyzing the association between DNA methylation and children’s IQ at age 6, including children’s age at follow-up, children’s BMI, maternal age during pregnancy, maternal education level, exposure to ETS at age 2, maternal IQ, and children’s sex.” This is a huge number of covariates for only n=59 samples. Were all of these included in each model? If so, I would have concerns about overfitting. Generally, you want at least 10-15 observations per term in a regression model. This is not the case for the 8 terms listed here. The authors should consider rerunning their analyses with a maximum of 5 or 6 terms, using selection criteria (e.g. AIC values) to select the best model.
Response 6: Thank you for your comment. We reanalyzed with selected 5 covariates including sex, maternal education level, exposure to ETS, maternal age, and maternal IQ after comparing AIC values in the model as follows. We changed the result in Section 2.6 as follows. We tested the association between the beta values for DNA methylation at these sites, and compared AIC among models. According to the model with the smallest AIC, we selected maternal age during pregnancy, exposure to ETS at age 2, maternal IQ, children’s sex as the covariates. As your comments, we showed the association between a significant CpG site and IQ scores instead of non-significant associations for the main table as follows.
|
IQ |
CpG sites |
Chr |
Gene |
Gene group |
Difference (95% CI)§ |
P-value* |
AIC |
|
Total IQ |
cg26269038 |
5 |
SLC6A53 |
Body |
3.66(-1.02, 8.34) |
0.126 |
443.79 |
|
Verbal IQ |
-0.56(-2.38, 1.27) |
0.550 |
340.17 |
||||
|
Performance IQ |
2.94(1.22, 4.66) |
0.001 |
333.47 |
Model before revision:
Model with reduced number of covariates:
1) selected maternal age during pregnancy, maternal education level, exposure to ETS at age 2, maternal IQ, children’s sex as the covariates.
|
IQ |
CpG sites |
Chr |
Gene |
Gene group |
Difference (95% CI)§ |
P-value* |
AIC |
|
Total IQ |
cg26269038 |
5 |
SLC6A3 |
Body |
3.17(-1.35, 7.68) |
0.170 |
441.17 |
|
Verbal IQ |
-0.82(-2.59, 0.95) |
0.365 |
338.04 |
||||
|
Performance IQ |
2.73(1.10, 4.35) |
0.001 |
328.66 |
2) selected maternal education level, exposure to ETS at age 2, maternal IQ, children’s sex as the covariates.
|
IQ |
CpG sites |
Chr |
Gene |
Gene group |
Difference (95% CI)§ |
P-value* |
AIC |
|
Total IQ |
cg26269038 |
5 |
SLC6A3 |
Body |
3.37(-1.32, 8.07) |
0.159 |
451.28 |
|
Verbal IQ |
-0.75(-2.56, 1.05) |
0.412 |
344.14 |
||||
|
Performance IQ |
2.80(1.10, 4.49) |
0.001 |
336.98 |
*3) selected maternal age during pregnancy, exposure to ETS at age 2, maternal IQ, children’s sex as the covariates.
|
IQ |
CpG sites |
Chr |
Gene |
Gene group |
Difference (95% CI)§ |
P-value* |
AIC |
|
Total IQ |
cg26269038 |
5 |
SLC6A3 |
Body |
3.68(-0.90, 8.26) |
0.115 |
440.41 |
|
Verbal IQ |
-0.66(-2.47, 1.15) |
0.475 |
338.13 |
||||
|
Performance IQ |
2.89(1.27, 4.51) |
0.001 |
326.12 |
|
Before revision |
|
Using the CpG sites significantly associated with exposure to greenness at age 2, we tested the association of these sites with total, verbal, and performance IQ scores at age 6, adjusting for children’s age at follow-up, children’s BMI, maternal age during pregnancy, maternal education level, exposure to ETS at age 2, maternal IQ, children’s sex, and cell type fractions. |
|
After revision |
|
Using the CpG sites significantly associated with exposure to greenness at age 2, we tested the association of DNA methylation levels at these CpG sites with total, verbal, and performance IQ scores at age 6 in multiple linear regression models, adjusting for maternal age during pregnancy, exposure to ETS at age 2, maternal IQ, and children’s sex using the selection criteria to select the best model by comparing the Akaike Information Criterion[43]. [43] Akaike, H. A new look at the statistical model identification. IEEE Trans Automat Contr 19, 716-723, http://doi.org/10.1109/TAC.1974.1100705 (1974). |
Point 7: Section 2.7: After filtering and processing steps, how many CpG sites were analyzed using the described modeling approach?
Response 7: Thank you for the comment. We elaborated more accurately about the process of filtering CpG sites in detail, showing how many CpG sites were finally analyzed. We changed the sentences as follows, and also modified Figure 1 as the following.
|
Before revision |
|
Method We also filtered CpG sites according to the following exclusion criteria: a) SNP-associated CpG sites defined as 0 or 1 base pair near SNP loci or minor allele frequency (MAF) > 5%; b) CpG sites that corresponded to the X or Y chromosome; c) CpG sites corresponding to non-CpG loci.
Result A total of 400 CpG sites were selected from 6 EWAS [26, 44-48]. Additionally, a total of 31,981 CpG sites selected, which were annotated to 835 genes reported from 13 GWAS after excluding duplicate genes [49-61]. These CpG sites were filtered according to the quality control method introduced earlier. As a result, 209 CpG sites from the EWAS and 8,534 CpG sites from the GWAS were finally selected (Table S2 and Table S3). |
|
After revision |
|
Method We also filtered CpG sites according to the following exclusion criteria: a) SNP-associated CpG sites defined as 0 or 1 base pair near SNP loci or minor allele frequency (MAF) > 5% (213,660 CpG sites); b) CpG sites that corresponded to the X or Y chromosome (19,627 CpG sites); c) CpG sites corresponding to non-CpG loci (3,627 CpG sites); d) cross-reactive probes (42,558 CpG sites). We were finally leaving 256,866 CpG sites to overlap the available data of 6 years-old for further analysis.
Result A total of 400 CpG sites were selected from 6 EWAS [26, 44-48]. Additionally, a total of 31,981 CpG sites selected, which were annotated to 835 genes reported from 13 GWAS after excluding duplicate genes [49-61]. As a result, 8,743 CpG sites were finally selected (Table S2 and Table S3). |
Point 8: Section 2.7: “Using the CpG sites significantly associated with exposure to greenness at age 2, we tested the association of these sites with total, verbal, and performance IQ scores at age 6, adjusting for children’s age at follow-up, children’s BMI, maternal age during pregnancy, maternal education level, exposure to ETS at age 2, maternal IQ, children’s sex, and cell type fractions.” I assume you tested the association between the beta values for DNA methylation at these sites with your predictors? If so, please make this clear in the text.
Response 8: Thank you for your comments. We rewrote the sentence more clearly as follows.
|
Before revision |
|
Using the CpG sites significantly associated with exposure to greenness at age 2, we tested the association of these sites with total, verbal, and performance IQ scores at age 6, adjusting for children’s age at follow-up, children’s BMI, maternal age during pregnancy, maternal education level, exposure to ETS at age 2, maternal IQ, children’s sex, and cell type fractions |
|
After revision |
|
Using the CpG sites significantly associated with exposure to greenness at age 2, we tested the association of DNA methylation levels at these CpG sites with total, verbal, and performance IQ scores at age 6 in multiple linear regression models, adjusting for maternal age during pregnancy, exposure to ETS at age 2, maternal IQ, and children’s sex using the selection criteria to select the best model by comparing the Akaike Information Criterion. |
Point 9: Table 2: “Change in DNA methylation by 1 interquartile range increases the proportions of greenness within the given radii from a residential address.” This statement is unclear. What does “by interquartile range increases the proportions of greenness” mean? Is this the change in DNA methylation by an increase of 1 in the interquartile range of greenness? And futher, are these percentages of DNA methylation or raw beta values? Please clarify by improving language here.
Response 9: Thank you for the comment. Residential greenness is measured in the percentage of green area within the circle with a given buffer radius (100m, 500m, 1000m etc). We estimated beta value of the association residential greenness and DNA methylation using linear regression. We showed the changes in DNA methylation per 1 IQR increase of the percentage of greenness within buffer radii. The levels of DNA methylation are given as raw beta value from the BMIQ normalized data. Beta value is given as the ratio of the methylated probe intensity and the overall intensity (sum of methylated and unmethylated probe intensities). We changed the footnote of Table 2 as follows.
|
Before revision |
|
§ Change in DNA methylation by 1 interquartile range increases the proportions of greenness within the given radii from a residential address. |
|
After revision |
|
§ Change in DNA methylation level by an increase of 1 interquartile range of greenness percentage within each buffer. |
Point 10: Table 2: What do the authors mean by a “robustly detected gene”? This is not explained anywhere in the text.
Response 10: Thank you for the point-out. We intended genes that were associated with residential greenness in more than two greenness types or more than two buffer radii. For example, The CpG site (cg00252813) in GAPDH was associated with residential greenness both in overall greenness and natural greenness. The CpG site (cg00809988) at ELAVL2 was associated with natural greenness both at 1000m and 2000m buffer radii. We deleted “robusted detected gene” to avoid confusion.
|
Before revision |
|
Abbreviations: CpG site location- based on Illumina annotation, derived from the University of California, Santa Cruz (UCSC), Adjusted for children’s age, children’s BMI, maternal education level, cell type fractions (CD8+T cells, CD4+T cells, natural killer cells, B cells, monocytes, and neutrophils), environmental tobacco smoke, maternal age, and children’s sex. The list was significantly associated as per Bonferroni correction (p < 0.05). * Indicated a robustly detected gene. † Analyzed using a linear regression model. § Change in DNA methylation by 1 interquartile range increases the proportions of greenness within the given radii from a residential address. |
|
After revision |
|
. Abbreviations: CpG site location- based on Illumina annotation, derived from the University of California, Santa Cruz (UCSC), Adjusted for children’s age, children’s BMI, maternal education level, cell type fractions (CD8+T cells, CD4+T cells, natural killer cells, B cells, monocytes, and neutrophils), environmental tobacco smoke, maternal age, and children’s sex. The list was significantly associated as per Bonferroni correction (p < 0.05). † Analyzed using a linear regression model. § Change in DNA methylation by 1 interquartile range increases the proportions of greenness within the given radii from a residential address |
Point 11: Table 2: A lot of the significant sites appear to show 0.000 as the change in DNA methylation by greenness. This suggests false positives could be occuring, possibly as a result of overfitting. Do the authors have an alternate explanation for these results?
Response 11: Thank you for your comments. We added the DNA methylation difference per IQR increase in each greenness exposure, and 95% confidence interval. In addition, we rechecked a mistake as follows.
|
Before revision |
|||||||||||||||||||||||||||||||||||||||||||||||||||||||||||||||||||||||||||||||||||||||||||||||||||||||||||||||||||||||||||||||||||||||||||||||||||||||||||||||||||||||||||||||||||||||||||||
|
|||||||||||||||||||||||||||||||||||||||||||||||||||||||||||||||||||||||||||||||||||||||||||||||||||||||||||||||||||||||||||||||||||||||||||||||||||||||||||||||||||||||||||||||||||||||||||||
|
After revision |
|||||||||||||||||||||||||||||||||||||||||||||||||||||||||||||||||||||||||||||||||||||||||||||||||||||||||||||||||||||||||||||||||||||||||||||||||||||||||||||||||||||||||||||||||||||||||||||
|
Point 12: Section 3.4: In the methods, the authors described using a Bonferroni-corrected P ≤ 0.05 cutoff for detection of significant pathways. It appears only the top pathway in Figure 3 reaches this cutoff, so including the other non-significant pathways in Figure 3.4 is rather misleading. Please either remove these pathways from the figure or clarify in the text that only a single pathway -- neurotransmitter clearance – was significant at the selected cutoff.
Response 12: Thank you for your comment. We did the functional enrichment pathway to identify the functional pathway between greenness and cognitive abilities-associated CpGs in literature reviews. We roughly set up the p-value as 0.1. Then, we found only a single pathway was significant at the selected cutoff value (p-value was 0.05). We added more explanation to clarify as follows.
|
Before revision |
|
We investigated potential biological functions by performing pathway enrichment analysis, which shows the top 20 pathways (Figure 2). Notably, the pathways of neuro-transmitter clearance associated with the SLC6A3 and SLC6A4 genes were significantly related to greenness exposure, of which SLC6A3 also showed significant associations with IQ in this study (adjusted p-value: 0.009; Table S4). In addition, the pathway enrichment analysis identified transmission across chemical synapses, opioid signaling, and neuronal systems pathway as playing an important role in the association between greenness and DNA methylation and contains genes associated with the nervous system (Figure 2). |
|
After revision |
|
We investigated potential biological functions by performing pathway enrichment analysis with the cutoff p-value set to 0.1. We found that the top 20 pathways including transmission across chemical synapses, opioid signaling, and neuronal systems pathway were observed (Figure 2A). Notably, a single pathway of neurotransmitter clearance was the only significantly enriched for the SLC6A3 and SLC6A4 genes at the selected cutoff (p-value was 0.05) (Table S4). SLC6A3 and SLC6A4 genes were significantly related to greenness exposure, of which SLC6A3 also showed significant associations with IQ in this study. Figure 2B showed linkages of the genes and biological concepts as network. In addition to neurotransmitter clearance pathway, SLC6A4 and SLC6A3 were non-significantly linked via transmission across chemical synapses (adjusted p-value: 0.12, respectively; Table S4) |
Point 13: Figure 2: The authors do not provide any description of Figure 3B in the text. Please provide language to assist readers with interpretation of this plot.
Response 13: Thank you for your comment. I am sure you mean Figure 2B instead of Figure 3B. We reorganized and rewrote this section as follows.
|
Before revision |
|
We investigated potential biological functions by performing pathway enrichment analysis, which shows the top 20 pathways (Figure 2). Notably, the pathways of neuro-transmitter clearance associated with the SLC6A3 and SLC6A4 genes were significantly related to greenness exposure, of which SLC6A3 also showed significant associations with IQ in this study (adjusted p-value: 0.009; Table S4). In addition, the pathway enrichment analysis identified transmission across chemical synapses, opioid signaling, and neuronal systems pathway as playing an important role in the association between greenness and DNA methylation and contains genes associated with the nervous system (Figure 2). |
|
After revision |
|
We investigated potential biological functions by performing pathway enrichment analysis with the cutoff p-value set to 0.1. We found that the top 20 pathways including transmission across chemical synapses, opioid signaling, and neuronal systems pathway were observed (Figure 2A). Notably, a single pathway of neurotransmitter clearance was the only significantly enriched for the SLC6A3 and SLC6A4 genes (adjusted p-value: 0.009; Table S4). SLC6A3 and SLC6A4 genes were significantly related to greenness exposure, of which SLC6A3 also showed significant associations with IQ in this study. Figure 2B showed linkages of the genes and biological concepts as network. In addition to neurotransmitter clearance pathway, SLC6A4 and SLC6A3 were non-significantly linked via transmission across chemical synapses (adjusted p-value: 0.12, respectively; Table S4) |
Point 14: Section 3.5: The authors do not provide clear language describing the fact that they re-analyzed the 25 CpG sites that were significant in the greenness models in new models using IQ. As such, the choice feels a bit arbitrary, especially since almost none of the CpG sites show a significant association with IQ. This is a lot of negative data for such a large table; consider better describing the results in the text and limiting data table to only sites that show significance in the follow-up, IQ model.
Response 14: Thank you for your comment. We added the results about the associations between the significant gene (SLC6A3) and total, verbal, and performance IQ as follows. In addition, we added the result which shows the association between 25 CpG sites and children’s IQ in Supplementary Table.
|
Before revision |
|
|
|
After revision |
|
|
Point 15: Figure 3: The x-axis label on this chart is unclear. What does “Exposure to total greenness in 100 m buffer at aged 2” referring to? Isn’t this a graph of quartiles? If so, please clarify the axis label to better explain the units.
Response 15: Thank you for your comment. We changed the x-axis as follows.
|
Before revision |
|
|
|
After revision |
|
|
Point 16: Discussion: “Recently, a review article reported that disruption in the clearance of neurotransmitters and increased amyloid β and tau from astrocytes appeared to be involved in neurotoxicity in Alzheimer’s disease patients”. The authors include this information, but then do little to build upon it. How does this relate to the results in the discussion?
Response 16: Thank you for your comment. As you comment, we would better delete the sentence in the discussion. Then, we re-arranged the discussion part to clarify the explanation of biological pathway between greenness and neurotransmitter-related genes as follows.
|
Before revision |
|
In our study, several genes significantly associated with greenness exposure, including PDE4D, PLCL1, GNG12, SLC6A4, and SLC6A3, were also linked to neurotransmitter clearance in the pathway enrichment analysis results. Signaling in the central nervous system is terminated by the clearance of neurotransmitters from the synapse via high-affinity transporter molecules in the presynaptic membrane [62]. Recently, a review article reported that disruption in the clearance of neurotransmitters and increased amyloid β and tau from astrocytes appeared to be involved in neurotoxicity in Alzheimer’s disease patients [63]. A CpG site at cg05016953 (SLC6A4) as a serotonin transporter, which was significantly associated with greenness exposure in our study, was reported to be regulated by a 5HTTLPR functional polymorphism, which was significantly associated with IQ scores in a previous study [64]. However, our results showed no significant association between DNA methylation changes at cg05016953 (SLC6A4) at age 2 and children’s IQ at age 6. Because there is a wide distribution of the 5HTTLPR genotype by race and ethnicity [65-67], further studies should be conducted among Asian children. |
|
After revision |
|
In our study, other several genes also significantly associated with greenness exposure, including PDE4D, PLCL1, GNG12, and SLC6A4 were also linked to neurotransmitter clearance in the pathway enrichment analysis results. Signaling in the central nervous system (CNS) is terminated by the clearance of neurotransmitters from the synapse via high-affinity transporter molecules in the presynaptic membrane [73]. Accumulated evidence has been shown that exposure to greenness has a positive effect on health by reducing oxidative stress [74-77]. Additionally, oxidative stress-induced damage to the brain is likely to negatively affect normal CNS functions [78]. Because these neurodegenerative disorders are related to increased environmental stressors, toxins, and oxidative stress in adults [79], brain development in children may also be linked to oxidative stress, which is reduced by greenness exposure. Similarly, oxidative stress is widely related to brain development. Recently, greenness exposure was significantly associated with reduced oxidative stress in Italian children [80]. We suggested that exposure to greenness, which was a pathway by reducing the oxidative stress, may be involved in neurodevelopment. [74] Yeager, Ray, et al. "Association between residential greenness and cardiovascular disease risk." Journal of the American Heart Association 7.24 (2018): e009117. [75] De Petris, Samuele, et al. "Geomatics and epidemiology: Associating oxidative stress and greenness in urban areas." Environmental Research 197 (2021): 110999. [76] Squillacioti, G., et al. "Greenness and physical activity as possible oxidative stress modulators in children." European Journal of Public Health 30.Supplement_5 (2020): ckaa165-090. [77] Squillacioti, G., et al. "Greenness effect on oxidative stress and respiratory flows in children." Environmental Epidemiology 3 (2019): 35-36. [78] Salim, Samina. "Oxidative stress and the central nervous system." Journal of Pharmacology and Experimental Therapeutics 360.1 (2017): 201-205. |
Point 17: Discussion: The discussion jumps between topics –DNA methyltransferases, greenness exposure, Parkinson’s disease, oxidative stress -- with little apparent follow-up or logical flow. The information presented is not wrong, but re-organization of the ideas into a clearer story is strongly suggested. For example, each paragraph could touch on one of these topics in depth, examining how the new results might play into what’s already known.
Response 17: Thank you for your comments. We re-organized the discussion section to clarify the ideas as follows.
|
Before revision |
|
In our study, several genes significantly associated with greenness exposure, including PDE4D, PLCL1, GNG12, SLC6A4, and SLC6A3, were also linked to neurotransmitter clearance in the pathway enrichment analysis results. Signaling in the central nervous system is terminated by the clearance of neurotransmitters from the synapse via high-affinity transporter molecules in the presynaptic membrane [62]. Recently, a review article reported that disruption in the clearance of neurotransmitters and increased amyloid β and tau from astrocytes appeared to be involved in neurotoxicity in Alzheimer’s disease patients [63]. A CpG site at cg05016953 (SLC6A4) as a serotonin transporter, which was significantly associated with greenness exposure in our study, was reported to be regulated by a 5HTTLPR functional polymorphism, which was significantly associated with IQ scores in a previous study [64]. However, our results showed no significant association between DNA methylation changes at cg05016953 (SLC6A4) at age 2 and children’s IQ at age 6. Because there is a wide distribution of the 5HTTLPR genotype by race and ethnicity [65-67], further studies should be conducted among Asian children. The most significant DNA methylation change at cg26269038 is located in the body, intron between the 3rd and 4th exon, of the gene solute carrier family 6, the member 3 (SLC6A3) on chromosome 5. The gene encodes a dopamine transporter (DAT), which is a member of the sodium- and chloride-dependent neurotransmitter transporter family, and provides rapid clearance of dopamine [68, 69], which mediates the reuptake of dopamine from the synaptic cleft [70]. Cómbita et al. (2017) determined whether SLC6A3/DAT1 gene contributed to individual differences in children’s self-regulation skills [71]. They evaluated self-regulation skills and cognitive tasks such as conflict processing, inhibitory control, and intelligence assessments in 127 children at ages 4 and 6 in Spain. They found that the presence of the 10 alleles of the SLC6A3 gene was related to a declining function of the dopaminergic transmission system, which was associated with poorer performance in self-regulation. Dopaminergic neurotransmission related to the SLC6A3 and DRD2 genes is reportedly associated with cognitive capacities, such as IQ, in previous studies [72-74]. Further study should be investigated that epigenetic regulation of gene expression is modulated by environmental factors such as exposure to stress and level of physical activity [72, 75, 76]. Epigenetic markers, such as DNA methylation, are dynamically reprogrammed during gametogenesis and early embryo preimplantation [77, 78]. Experimental evidence suggests that the epigenome of mammalian embryonic cells is more susceptible to environmental stimulation than other differentiated cells [18, 79] because of the abundance of de novo DNA methyltransferases in these rapidly dividing pluripotent cells [77, 78]. In line with previous findings, we found that greenness exposure in early childhood is a modifiable factor related to DNA methylation change, which was found to be associated with cognitive ability in a previous study. The SLC6A3 gene, which accumulates cytotoxic dopamine or other toxins in dopamine neurons, is a risk factor for Parkinson’s disease [80]. Because these neurodegenerative disorders are related to increased environmental stressors, toxins, and oxidative stress in adults [81], brain development in children may also be linked to oxidative stress, which is reduced by greenness exposure. Similarly, oxidative stress is widely related to brain development. Recently, greenness exposure was significantly associated with reduced oxidative stress in Italian children [82]. Our results show that children with higher greenness exposure had lower DNA methylation levels of the SLC6A3 gene. This region might be linked to greenness exposure and neurological development in children. However, further studies are needed to understand how these cognitive ability-related CpG sites are linked to greenness exposure. |
|
After revision |
|
Epigenetic markers, such as DNA methylation, are dynamically reprogrammed during gametogenesis and early embryo preimplantation [62, 63]. Experimental evidence suggests that the epigenome of mammalian embryonic cells is more susceptible to environmental stimulation than other differentiated cells [18, 64] because of the abundance of de novo DNA methyltransferases in these rapidly dividing pluripotent cells [65, 66]. The most significant DNA methylation change at cg26269038 is located in the body, intron between the 3rd and 4th exon, of the gene solute carrier family 6, the member 3 (SLC6A3) on chromosome 5. The gene encodes a dopamine transporter (DAT), which is a member of the sodium- and chloride-dependent neurotransmitter transporter family, and provides rapid clearance of dopamine [67, 68], which mediates the reuptake of dopamine from the synaptic cleft [69]. Cómbita et al. (2017) determined whether SLC6A3/DAT1 gene contributed to individual differences in children’s self-regulation skills [70]. They evaluated self-regulation skills and cognitive tasks such as conflict processing, inhibitory control, and intelligence assessments in 127 children at ages 4 and 6 in Spain. They found that the presence of the 10 alleles of the SLC6A3 gene was related to a declining function of the dopaminergic transmission system, which was associated with poorer performance in self-regulation. Dopaminergic neurotransmission related to the SLC6A3 and DRD2 genes is reportedly associated with cognitive capacities, such as IQ, in previous studies [71-73]. Our results show that children with higher greenness exposure had lower DNA methylation levels of the SLC6A3 gene. This region might be linked to greenness exposure and neurological development in children. However, further studies are needed to understand how these cognitive ability-related CpG sites are linked to greenness exposure. In line with previous findings, we found that greenness exposure in early childhood is a modifiable factor related to DNA methylation change, which was found to be associated with cognitive ability in a previous study. In our study, other several genes also significantly associated with greenness exposure, including PDE4D, PLCL1, GNG12, and SLC6A4, were also linked to neurotransmitter clearance in the pathway enrichment analysis results. Signaling in the central nervous system (CNS) is terminated by the clearance of neurotransmitters from the synapse via high-affinity transporter molecules in the presynaptic membrane [73]. Accumulated evidence has been shown that exposure to greenness has a positive effect on health by reducing oxidative stress [74-77]. Additionally, oxidative stress-induced damage to the brain is likely to negatively affect normal CNS functions [78]. Because these neurodegenerative disorders are related to increased environmental stressors, toxins, and oxidative stress in adults [79], brain development in children may also be linked to oxidative stress, which is reduced by greenness exposure. Similarly, oxidative stress is widely related to brain development. Recently, greenness exposure was significantly associated with reduced oxidative stress in Italian children [80]. We suggested that exposure to greenness, which was a pathway by reducing the oxidative stress, may be involved in neurodevelopment. A CpG site at cg05016953 (SLC6A4) as a serotonin transporter, which was significantly associated with greenness exposure in our study, was reported to be regulated by a 5HTTLPR functional polymorphism, which was significantly associated with IQ scores in a previous study [81]. However, our results showed no significant association between DNA methylation changes at cg05016953 (SLC6A4) at age 2 and children’s IQ at age 6. Because there is a wide distribution of the 5HTTLPR genotype by race and ethnicity [82-84], further studies should be conducted among Asian children. |
Point 18: Discussion: The authors do not provide a discussion of whether the effect sizes they see (e.g. ~1-2% change in DNAm at SLC6A3 gene) are expected to be functionally relevant. Would this level of DNA methylation change actually impact gene regulation?
Response 18: Thank you for your important question. The effect sizes of the association between residential greenness and DNA methylation were within 1~3% in our study. In an Australian study which investigated the association between greenness and epigenome-wide DNA methylation, the coefficients ranged from -0.36% to 1.73% (Xu et al, 2021). Epidemiological studies concerning the effects of environmental exposures typically shows small effect sizes. For example, the differences in DNA methylation reported between exposed vs. unexposed groups are generally on the scale of 2-10%, and in some cases even smaller differences have been observed (Breton et al, 2017). It has been reported that for every 1% change in methylation at the differentially methylated region at IGF2, a halving or doubling of IGF2 transcription was observed (Murphy et al, 2012). Although such a few percent change in DNA methylation appears as a small effect size, it is only so in the perspective of population of cells. At a single cell level, a CpG site is either methylated (100%), hemi-methylated (50%) or non-methylated (0%), and such a difference could have substantial effects in cell functions including gene regulation (Breton et al, 2017). We added this explanation in the discussion section as the following.
|
Before revision |
|
- |
|
After revision |
|
The effect sizes of the association between residential greenness and DNA methylation were within 1~3%. In an Australian study which investigated the association between greenness and epigenome-wide DNA methylation, the coefficients ranged from -0.36% to 1.73% [27]. Epidemiological studies concerning the effects of environmental exposures typically shows small effect sizes. For example, the differences in DNA methylation reported between exposed vs. unexposed groups are generally on the scale of 2-10%, and in some cases even smaller differences have been observed [85]. It has been reported that for every 1% change in methylation at the differentially methylated region at IGF2, a halving or doubling of IGF2 transcription was observed [86]. Although such a few percent change in DNA methylation appears as a small effect size, it is only so in the perspective of population of cells. At a single cell level, a CpG site is either methylated (100%), hemi-methylated (50%) or non-methylated (0%), and such a difference could have substantial effects in cell functions including gene regulation [85]. [85] Breton, Carrie V., et al. "Small-magnitude effect sizes in epigenetic end points are important in children’s environmental health studies: the children’s environmental health and disease prevention research center’s epigenetics working group." Environmental health perspectives 125.4 (2017): 511-526. [86] Murphy, Susan K., et al. "Gender-specific methylation differences in relation to prenatal exposure to cigarette smoke." Gene 494.1 (2012): 36-43. |
Reviewer 2 Report
I set out the following observations and considerations below:
- I suggest that the legibility of the figures should be improved, especially figure 1.
- At the bottom of Table 1 there is information about unused abbreviations such as: 3-PBA, 3-phenoxybenzoic acid; ADHD, attention-deficit/hyperactivity disorder; AI, inattention; HI, hyperactivity-impulsivity.
- Unobserved confounding variables that could be used in the analysis: mother's occupation, breastfeeding pattern, mother's previous smoking, pattern of child care in the period up to two years of age (nursery, at home, etc.). The limitations of the study have been addressed, but I think these can be considered.
- I assume that the aforementioned Institutional Review Board of Seoul National University College of Medicine (IRB No. 1201-010-392), in the article, plays the role of an Ethics Committee on Human Research. If so, I have not observed relevant limitations in the methodology.
Author Response
Thank you for providing us another opportunity to improve the quality of our study. We have further revised the manuscript according to the reviewers’ valuable and helpful comments, and have provided our point-by-point responses to each comment.
Point 1: I set out the following observations and considerations below: I suggest that the legibility of the figures should be improved, especially figure 1.
Response 1: Thank you for your comment. We changed the figure 1 to improve legibility.
|
Before revision |
|
|
|
After revision |
|
|
Point 2: Table 1 there is information about unused abbreviations such as: 3-PBA, 3-phenoxybenzoic acid; ADHD, attention-deficit/hyperactivity disorder; AI, inattention; HI, hyperactivity-impulsivity.
Response 2: Thank you for your comment. We deleted unused abbreviations as follows.
|
Before revision |
|
Abbreviations: EDC, Environment and the Development of Children; SD, standard deviation; ETS, environmental tobacco smoke; 3-PBA, 3-phenoxybenzoic acid; ADHD, attention-deficit/hyperactivity disorder; IA, inattention; HI, hyperactivity-impulsivity. |
|
After revision |
|
Abbreviations: EDC, Environment and the Development of Children; SD, standard deviation; ETS, environmental tobacco smoke; BMI, body mass index; IQ, intelligence quotient |
Point 3: 3. Unobserved confounding variables that could be used in the analysis: mother's occupation, breastfeeding pattern, mother's previous smoking, pattern of child care in the period up to two years of age (nursery, at home, etc.). The limitations of the study have been addressed, but I think these can be considered.
Response 3: Thank you for your comments. According to the other reviewer’s opinion, we have too many covariates in the model with a small sample size. Therefore, we found the best fit model using Akaike Information Criterion to test the association between DNA methylation levels and children’s IQ. The result showed that the model, which was adjusted for maternal age, children’s sex, maternal IQ, and exposure to ETS at age 2, showed the smallest AIC value. Thus, we changed the covariates as follows. In addition, we provided the result with additional available covariates such as breastfeeding pattern, and mother’s previous smoking status and included it in Supplementary Table S6 as follows.
|
IQ |
CpG sites |
Chr |
Gene |
Gene group |
Difference (95% CI)§ |
P-value* |
AIC |
|
Total IQ |
cg26269038 |
5 |
SLC6A53 |
Body |
3.66(-1.02, 8.34) |
0.126 |
443.79 |
|
Verbal IQ |
-0.56(-2.38, 1.27) |
0.550 |
340.17 |
||||
|
Performance IQ |
2.94(1.22, 4.66) |
0.001 |
333.47 |
Before revision model:
Reduced covariates models:
1) selected maternal age during pregnancy, maternal education level, exposure to ETS at age 2, maternal IQ, children’s sex as the covariates.
|
IQ |
CpG sites |
Chr |
Gene |
Gene group |
Difference (95% CI)§ |
P-value* |
AIC |
|
Total IQ |
cg26269038 |
5 |
SLC6A3 |
Body |
3.17(-1.35, 7.68) |
0.170 |
441.17 |
|
Verbal IQ |
-0.82(-2.59, 0.95) |
0.365 |
338.04 |
||||
|
Performance IQ |
2.73(1.10, 4.35) |
0.001 |
328.66 |
2) selected maternal education level, exposure to ETS at age 2, maternal IQ, children’s sex as the covariates.
|
IQ |
CpG sites |
Chr |
Gene |
Gene group |
Difference (95% CI)§ |
P-value* |
AIC |
|
Total IQ |
cg26269038 |
5 |
SLC6A3 |
Body |
3.37(-1.32, 8.07) |
0.159 |
451.28 |
|
Verbal IQ |
-0.75(-2.56, 1.05) |
0.412 |
344.14 |
||||
|
Performance IQ |
2.80(1.10, 4.49) |
0.001 |
336.98 |
*3) selected maternal age during pregnancy, exposure to ETS at age 2, maternal IQ, children’s sex as the covariates.
|
IQ |
CpG sites |
Chr |
Gene |
Gene group |
Difference (95% CI)§ |
P-value* |
AIC |
|
Total IQ |
cg26269038 |
5 |
SLC6A3 |
Body |
3.68(-0.90, 8.26) |
0.115 |
440.41 |
|
Verbal IQ |
-0.66(-2.47, 1.15) |
0.475 |
338.13 |
||||
|
Performance IQ |
2.89(1.27, 4.51) |
0.001 |
326.12 |
4) selected maternal education level, exposure to ETS at age 2, maternal IQ, children’s sex, breastfeeding status, and mother's previous smoking status as the covariates.
|
IQ |
CpG sites |
Chr |
Gene |
Gene group |
Difference (95% CI)§ |
P-value* |
AIC |
|
Total IQ |
cg26269038 |
5 |
SLC6A3 |
Body |
2.60(-1.73, 6.92) |
0.240 |
438.46 |
|
Verbal IQ |
-1.05(-2.77, 0.66) |
0.228 |
336.76 |
||||
|
Performance IQ |
2.61(1.02, 4.20) |
0.001 |
328.07 |
4) selected maternal education level, exposure to ETS at age 2, maternal IQ, children’s sex, and mother's previous smoking status as the covariates.
|
IQ |
CpG sites |
Chr |
Gene |
Gene group |
Difference (95% CI)§ |
P-value* |
AIC |
|
Total IQ |
cg26269038 |
5 |
SLC6A3 |
Body |
3.27(-1.25, 7.79) |
0.157 |
442.14 |
|
Verbal IQ |
-0.78(-2.58, 1.03) |
0.399 |
341.00 |
||||
|
Performance IQ |
2.74(1.14, 4.34) |
0.001 |
327.78 |
|
Before revision |
|
|
|
After revision |
|
Result In sensitive analysis, we analyzed these association with additional covariates such as breastfeeding pattern, and mother's previous smoking status in Table S6. The result was not different from the main result in Table 3. |
Point 4: 4. I assume that the aforementioned Institutional Review Board of Seoul National University College of Medicine (IRB No. 1201-010-392), in the article, plays the role of an Ethics Committee on Human Research. If so, I have not observed relevant limitations in the methodology.
Response 4: Thank you for your comment. Our study protocol was approved by the Institutional Review Board of Seoul National University College of Medicine (IRB No. 1201-010-392) including Ethics approval and consent to participants. Although we mentioned it at the end of the manuscript, we added the sentence in the method section as follows.
|
Before revision |
|
Institutional Review Board Statement: The study protocol was approved by the Institutional Review Board of Seoul National University College of Medicine (IRB No. 1201-010-392). 2. Materials and Methods 2.1. Study population Our research was based on a subset of the EDC study cohort, an ongoing prospective cohort study designed to evaluate the association between prenatal and postnatal environmental exposures and physical or cognitive development. Detailed information on the study design has been described elsewhere [32]. Briefly, a total of 726 eligible pregnant women from eight local hospitals in Seoul and Gyeonggi province of South Korea were enrolled from August 2008 to July 2010. We collected urine and blood samples to estimate exposure to environmental factors during the second trimester of pregnancy. A total of 425 children aged 2 years and 574 children aged 6 years at enrolment were followed up. DNA methylation analysis was conducted in a sub-study of 59 participants using blood samples collected at the age of 2 years.
|
|
After revision |
|
Institutional Review Board Statement: The study protocol was approved by the Institutional Review Board of Seoul National University College of Medicine (IRB No. 1201-010-392). 2. Materials and Methods 2.1. Study population Our research was based on a subset of the EDC study cohort, an ongoing prospective cohort study designed to evaluate the association between prenatal and postnatal environmental exposures and physical or cognitive development. Detailed information on the study design has been described elsewhere [32]. Briefly, a total of 726 eligible pregnant women from eight local hospitals in Seoul and Gyeonggi province of South Korea were enrolled from August 2008 to July 2010. We collected urine and blood samples to estimate exposure to environmental factors during the second trimester of pregnancy. A total of 425 children aged 2 years and 574 children aged 6 years at enrolment were followed up. DNA methylation analysis was conducted in a sub-study of 59 participants using blood samples collected at the age of 2 years. The study protocol including ethical approval and participant consent was reviewed and approved by the Institutional Review Board of the Seoul National University Hospital (IRB No. C-1201-010-392). |
Reviewer 3 Report
An interesting and well-conducted manuscript that carries a significant message to parents and decision makes. The manuscript also highlights the importance of children’s health.
In results:
The sentence “The mean age and BMI of children were 23.32 months (SD: 0.77 months) and 16.57 kg/m2 (SD: 1.20 kg/m2), respectively“ repetitive sentence.
Did you included only 30 girls and 29 boys, right? „ There were similar numbers of girls and boys in the study (30 and 29, respectively).“
I did not find a clearly defined novelty in the article. Maybe it could be made more noticeable?
Author Response
Thank you for providing us another opportunity to improve the quality of our study. We have further revised the manuscript according to the reviewers’ valuable and helpful comments, and have provided our point-by-point responses to each comment.
Point 1: An interesting and well-conducted manuscript that carries a significant message to parents and decision makes. The manuscript also highlights the importance of children’s health.
In results:
The sentence “The mean age and BMI of children were 23.32 months (SD: 0.77 months) and 16.57 kg/m2 (SD: 1.20 kg/m2), respectively“ repetitive sentence.
Response 1: Thank you for your comment. We deleted the sentence to avoid the repetition.
|
Before revision |
|
The mean age and BMI of children were 23.32 months (SD: 0.77 months) and 16.57 kg/m2 (SD: 1.20 kg/m2), respectively. The mean age and BMI of children were 23.32 months (SD: 0.77) and 16.57 kg/m2 (SD: 1.20 kg/m2), respectively. |
|
After revision |
|
The mean age and BMI of children were 23.32 months (SD: 0.77) and 16.57 kg/m2 (SD: 1.20 kg/m2), respectively. |
Point 2: Did you included only 30 girls and 29 boys, right? „ There were similar numbers of girls and boys in the study (30 and 29, respectively).“
Response 2: Thank you for your comment. As you mentioned, the numbers of girls and boys were similar (50% vs 49%), and the total cohort has also similar numbers of girls and boys, which is indicated as 46% vs. 53% in Table 1.
Point 3: I did not find a clearly defined novelty in the article. Maybe it could be made more noticeable?
Response 3: Thank you for your comment. To our knowledge, this is the first epigenetic study showing the association between greenness exposure and cognitive ability-related DNA methylation changes in children. In addition, there were previously fewer studies exploring the biological mechanisms underlying the association between greenness exposure and desirable health effects. Therefore, we attempted to explain the biological pathways for the association between greenness and cognitive development in children. As in the highlighted part, we re-arranged the strength part as follows.
|
Before revision |
|
Our study had several strengths. First, we estimated the association between greenness exposure related to DNA methylation at age 2 and children’s IQ at age 6 in a prospective cohort study. To our knowledge, this is the first epigenetic study of the association between greenness exposure and cognitive ability-related DNA methylation changes in children, which might provide a clue to explain the causal role of greenness in neurodevelopment in children. Second, we estimated the proportions of greenness in buffers of various sizes and exposure to various types of greenness. There is currently insufficient evidence to determine which type of greenness exposure and which buffer size have the greatest impact on mental health in children, and further studies should be performed using buffers of various sizes and various types of greenness. |
|
After revision |
|
Our study had several strengths. First, to our knowledge, this is the first epigenetic study of the association between greenness exposure and cognitive ability-related DNA methylation changes in children. Second, we estimated the association between greenness exposure related to DNA methylation at age 2 and children’s IQ at age 6 in a prospective cohort study, which might provide a clue to explain the causal role of greenness in neurodevelopment in children. Third, we estimated the proportions of greenness in buffers of various sizes and exposure to various types of greenness. There is currently insufficient evidence to determine which type of greenness exposure and which buffer size have the greatest impact on mental health in children, and further studies should be performed using buffers of various sizes and various types of greenness. |
Round 2
Reviewer 1 Report
Thank you for your thorough response. You addressed all of my concerns in the revised manuscript.